# Interaction of *Trichoderma* Species with *Fusarium graminearum* Growth and Its Trichothecene Biosynthesis as Further Contribution in Selection of Potential Biocontrol Agents

**DOI:** 10.3390/jof11070521

**Published:** 2025-07-14

**Authors:** Xianfeng Ren, Lixia Fan, Guidong Li, Ilya V. Lyagin, Bingchun Zhang, Mingxiao Ning, Mengmeng Yan, Jing Gao, Fei Wang, Changying Guo, Antonio F. Logrieco

**Affiliations:** 1Institute of Agricultural Quality Standards and Testing Technology of Shandong Academy of Agricultural Sciences, Jinan 250100, China; renxianfenga@163.com (X.R.); superdemeter@163.com (L.F.); lguidong123@163.com (G.L.); llzbest66@163.com (B.Z.); mxning428@163.com (M.N.); yanmengmeng_10@163.com (M.Y.); 2Shandong Provincial Key Laboratory of Test Technology on Food Quality and Safety, Jinan 250100, China; 3College of Food and Bioengineering, Henan University of Science and Technology, Luoyang 471000, China; 4Faculty of Chemistry, Lomonosov Moscow State University, Moscow 119991, Russia; ilyagin@enzyme.chem.msu.ru (I.V.L.); faye9393@163.com (F.W.); 5Xianghu Laboratory (Zhejiang Agricultural Laboratory), Hangzhou 311200, China; gjing1997@126.com

**Keywords:** *Trichoderma*, *Fusarium graminearum*, deoxynivalenol, biocontrol

## Abstract

The interactions of *Fusarium graminearum* PG-Fg1 and its main trichothecenes with the 28 *Trichoderma* isolates were studied in vitro. The antagonistic effect assessed by dual-culture tests showed that *Trichoderma* isolates arrested the growth of PG-Fg1 after contact, overgrew the PG-Fg1 colony and inhibited the production of deoxynivalenol (DON), 3-acetyl-deoxynivalenol (3-ADON), and 15-acetyldeoxynivalenol (15-ADON) by up to 95.3%, 99.4%, and 99.6%, respectively. PG-Fg1 was hard to overgrow *Trichoderma* for further extension. Additionally, the inhibitory effects on PG-Fg1 by the *Trichoderma* metabolites, including volatiles and non-volatiles, were also investigated. Most of the *Trichoderma* isolates produced metabolites which inhibited PG-Fg1 growth and mycotoxins production. Specifically, *Trichoderma* non-volatiles and volatiles showed *Fusarium* growth inhibition rates ranging from 7% to 72% and 3% to 32%, respectively. Notably, non-volatile compounds from two isolates and volatiles from one isolate up-regulated the expression of DON biosynthesis genes (tri4 and tri5), leading to increased production of DON, 3-ADON, and 15-ADON. This study highlights the potential risk posed by certain *Trichoderma* strains as microbial agents, which can stimulate toxigenic fungi to produce higher levels of mycotoxins. Based on our results and previous reports, when selecting *Trichoderma* species as biocontrol agents against *Fusarium graminearum*, its effects on mycotoxins production should be carefully assessed, particularly given observed stimulatory impacts.

## 1. Introduction

The toxigenic fungi, including *Fusarium*, *Aspergillus*, *Penicillium*, and *Alternaria*, are capable of producing mycotoxins, which are toxic secondary metabolites commonly detected in foodstuffs and feeds [1,2]. Among these, *Fusarium* species, one of the dominant toxigenic fungal genera, comprising over 300 phylogenetically distinct species, are particularly significant, as they commonly infect crops pre- or post-harvest [3], leading to reduced yields and diminished crop quality. Fusarium head blight (FHB), a disease caused by *Fusarium* species [4] in wheat, is notably destructive. *Fusarium graminearum* has been identified as the predominant causative agent of FHB and is frequently isolated from infected wheat globally [5]. In fact, *F. graminearum* was ranked fourth among the top ten fungal pathogens in a 2012 survey [6]. This pathogen attacks wheat spikes, compromising grain production and quality through the production of two major classes of mycotoxins: type B trichothecenes, such as deoxynivalenol (DON), 3-acetyl-deoxynivalenol (3-ADON), 15-acetyl-deoxynivalenol (15-ADON), nivalenol (NIV), Fusarennon-X (FUS), and estrogenic zearalenone (ZEN) [7,8,9,10,11]. Among the diverse mycotoxins produced by *F. graminearum*, DON and its derivatives 3-ADON and 15-ADON are particularly persistent and represent some of the most frequently detected contaminants in grains such as maize, wheat, barley, oat, and their products [7]. A higher dose of DON causes significant reduction in body weight, severe damage to hematopoietic systems, and immune dysregulation in animals [1]. DON raises great public health concerns due to its high prevalence, and their teratogenic, mutagenic, and immunosuppressive effects. Therefore, the development of efficient strategies to control *F. graminearum* and DON contamination is critically important.

Various microbial agents have been evaluated for the control of *F. graminearum* and DON contamination, including bacteria of the genera *Bacillus* [12,13,14], *Pseudomonas* [15,16], *Streptomyces* [17,18], and *Lysobacter* [19,20] (Kim et al., 2019; Odhiambo et al., 2017), as well as endophytic fungi belonging to genera *Alternaria*, *Chaetomium*, *Clonostachys*, *Curvularia*, *Epicoccum*, *Penicillium*, and *Piriformospora* [21,22]. Notably, *Trichoderma* spp., a well-known group of biocontrol fungi, commonly found in soil and root ecosystems, are particularly effective in controlling plant diseases [23,24,25]. Extensive research has demonstrated that *Trichoderma* exerts multiple beneficial effects on plant growth, resilience, and yield [26,27]. Therefore, *Trichoderma* has become a shining star as a microbial agent, characterized by its wide availability, biosafety, high efficiency, and universal functions.

As reviewed by Benítez et al. [28], *Trichoderma* strains exhibit biocontrol against pathogens through indirect mechanisms, such as competing for space and nutrients, antagonizing pathogens, biofertilization, stimulating plant defense, modifying the rhizosphere, and producing antimicrobial compounds, or through direct mechanisms, such as mycoparasitism. It is widely believed that multiple mechanisms govern the biocontrol of pathogens by *Trichoderma* spp. Previous research on the interaction between *Trichoderma* and *Fusarium* has primarily focused on competition [29,30], antagonistic activity [31,32], and the production of antibiotics or enzymes to inhibit mycelial growth [33,34]. These studies have explored various aspects of microbial interaction, including fungi–plant interaction and *Trichoderma*–pathogen interaction [35,36]. However, less attention has been given to mycotoxin analysis. Consequently, the interactions among *Trichoderma*, toxigenic fungi, and mycotoxins have not been clearly elucidated.

This study (a) examined the antagonist interactions between the *F. graminearum* strain producing trichothecenes (DON, 3-ADON, and 15A-DON) and various *Trichoderma* spp., (b) evaluated the inhibitory activity of volatiles and non-volatiles produced by the *Trichoderma* spp. on *F. graminearum* growth and mycotoxin biosynthesis, (c) assessed the tolerance of *Trichoderma* to DON, 3-ADON, and 15A-DON, and (d) tried to elucidate the underlying mechanisms of *Trichoderma* against *F. graminearum* and its trichothecenes. Through this comprehensives tudy, we analyzed the potential of *Trichoderma* spp. for biocontrol of *F. graminearum* and the reduction of trichothecenes through different mechanisms, and particularly emphasized the hazardous effects of certain *Trichoderma* strains as antimicrobial agents, thereby offering new insights into the use of *Trichoderma* as biocontrol agents.

## 2. Materials and Methods

### 2.1. Chemicals

The chemical standards of mycotoxins (purity > 99%) were purchased from Sigma-Aldrich (St. Louis, MO, USA). Methanol, acetonitrile, and formic acid (LC-MS grade) were purchased from Macklin Biochemical Co., Ltd. (Shanghai, China). A nylon syringe filter (13 mm, 0.22 µm) was obtained from Jiangsu Green Union Science Instrument Co., Ltd. (Nanjing, China). The water was purified by a Milli-Q purification system (Millipore, Bedford, MA, USA). Potato dextrose agar (PDA) and potato dextrose broth (PDB) were purchased from Hopebio Co., Ltd. (Qingdao, China) and used for the solid and liquid cultures, respectively, of *Trichoderma* and *F. graminearum*.

### 2.2. Fungal Strains

The 28 *Trichoderma* strains used in this study were collected from various substrates (soil, decaying wood, or plant debris) originating from Tibet, Shandong, and Yunnan Province in China, and were coded and stored according to Laboratory protocol at Shandong Academy of Agricultural Sciences in Jinan, China (Appendix A). To ensure the purity of the *Trichoderma* strains, conidia were resuspended in sterile distilled water and inoculated at a low density on 90 mm water agar (WA) Petri dishes. After 18–24 h of incubation at 25 °C in the dark, a single germinated conidium was transferred to 60 mm PDA plates using a dissecting microscope. All the strains were molecularly identified via sequence analysis of the internal transcribed spacer regions ITS-1 and ITS-2 of the nuclear rDNA as reported by Fanelli et al. [37]. Additionally, all strains were identified morphologically, according to Gams and Bisset [38]. *Fusarium graminearum* PG-Fg1 was originally isolated from wheat in Jinan, China, and is a high DON-, 3-ADON-, and 15-ADON-producing strain. Fungal cultures were maintained in purity on potato dextrose agar slants at 4 °C, which were used as sources of inocula for subsequent fresh cultures.

### 2.3. Antagonism of Trichoderma spp. Against F. graminearum Growth and DON/3-ADON/15-ADON Production

The antagonism and colony interaction between each *Trichoderma* strain and *F. graminearum* PG-Fg1 were investigated in vitro using dual cultures. In a 9 cm diameter Petri dish containing 20 mL of potato dextrose agar (PDA), a 6 mm diameter mycelial plug from a fresh *Trichoderma* culture was inoculated 1 cm from the edge of the dish, while a 6 mm diameter mycelial plug of *Fusarium graminearum* PG-Fg1 was inoculated on the other side of the dish at a distance of 7 cm from the *Trichoderma* inoculation point. All pathogen–antagonist co-cultures were incubated at 28 °C in the dark. The radial growth was measured daily until colony contact or cessation of growth if contact did not occur.

After 21 days of co-cultivation, the colony interactions were visually assessed and classified according to the criteria established by Whipps [39] as follows: 1 = *Trichoderma* overgrowing PG-Fg1 and PG-Fg1 stopped; 1/2 = *Trichoderma* overgrowing PG-Fg1 but PG-Fg1 still growing; 2 = PG-Fg1 overgrowing *Trichoderma* and *Trichoderma* stopped; 2/1 = PG-Fg1 overgrowing *Trichoderma* but *Trichoderma* still growing; 3 = slight mutual inhibition (inhibition zone ≤ 2 mm wide); 4 = strong mutual inhibition (inhibition zone > 2 mm and ≤ 4 mm-wide). The experiment was carried out in triplicate.

After 7 days of co-cultivation, mycelial plugs were excised at regular intervals along the radius of the PG-Fg1 colony using a 10 mm diameter cork-borer, starting from the inoculation point to the edge of the colony. Approximately 1 g of the sample was precisely weighed, transferred to test tubes, and extracted with 5 mL of an acetonitrile–water (80:20, *v*/*v*) solution on a shaker at 2000 rpm for 10 min at room temperature. The sample was then centrifuged, filtered, and diluted to obtain extracts [40], which were stored at −20 °C until the analysis of DON, 3-ADON, and 15-ADON. The inhibition rate of DON, 3-ADON, and 15-ADON production (%Ia) was calculated with the following formula%Ia = (A_C_ − A_T_)/A_C_ × 100(1)
where A_T_ was the amount of DON, 3-ADON, or 15-ADON produced by *F. graminearum* on medium mixed with *Trichoderma* metabolites, and A_C_ was the amount of DON, 3-ADON, or 15-ADON in the control plates.

### 2.4. Effect of Non-Volatiles of Trichoderma spp. on F. graminearum Growth and DON/3-ADON/15-ADON Production

The culture supernatant of *Trichoderma*, containing non-volatile metabolites, was prepared as described in our previous reports (Yue et al. 2022 [41]) with some modification. Five 6 mm diameter *Trichoderma* plugs were added to 100 mL PDB in the flask and incubated for 7 days with constant shaking (175 rpm) at 28 ± 1 °C. The culturing broth was collected and centrifuged at 15,000× *g* for 5 min to remove *Trichoderma* mycelium. The supernatant was subsequently filtered through filter paper and a 0.22 μm filter membrane, and then stored at 4 °C for the subsequent use.

The inhibitory effect of non-volatile metabolites of *Trichoderma* spp. on the growth of *F. graminearum* PG-Fg1 was studied in PDA. For the preparation of PDA media mixed with *Trichoderma* metabolites, 15 mL of PDA was autoclaved at 121 °C for 30 min. After cooling down to 55 °C, 5 mL of the *Trichoderma* culture supernatant was added to the liquid PDA, which was thoroughly mixed using a vortex mixer and quickly poured into 9 cm diameter Petri dishes [42]. After the media solidification, a 6 mm mycelial plug from the edge of a 7-day-old *F. graminearum* PG-Fg1 culture was placed at the center of the dish. The cultures were incubated at 28 °C in the dark for 7 days. A control dish was prepared by inoculating *F. graminearum* PG-Fg1 on standard PDA without *Trichoderma* metabolites. The experiment was carried out in triplicate.

The colony diameter of *F. graminearum* PG-Fg1 was measured with a ruler under a dissecting microscope every day. The growth inhibition of *F. graminearum* PG-Fg1 (%I_b_) caused by *Trichoderma* non-volatiles was calculated with the following formula%I_b_ = (B_C_ − B_T_)/B_C_ × 100%(2)
where B_T_ was the average diameter of *F. graminearum* PG-Fg1 colonies grown in treatment conditions, and B_C_ was the average diameter of *F. graminearum* PG-Fg1 colony grown in control conditions.

Meanwhile, after 7 days of growth, mycelial plugs were excised at regular intervals along the radius of the PG-Fg1 colony using a 10 mm diameter cork-borer, starting from the inoculation point to the edge of the colony. Approximately 1 g of sample was precisely weighed for the analysis of DON, 3-ADON, and 15-ADON as described in Section 2.3. The inhibition rate of DON, 3-ADON, and 15-ADON production was calculated with the formula of %Ia.

### 2.5. Effect of Non-Volatiles of Trichoderma GC-T88 on the Expression of DON Biosynthesis Genes

According to the above experiments, *Trichoderma* strains GC-T8 and GC-T88 increased the production of DON, 3-ADON, and 15-ADON by *F. graminearum*. To further understand the mechanism behind the result, we assessed the effect of GC-T88 non-volatiles on the expression of DON biosynthesis genes (*tri4*, *tri5*, *tri6*, *tri10*). After 5 days of incubation of *F. graminearum* PG-Fg1 in the presence of GC-T88 non-volatiles, the mycelia were collected by scraping them off the medium surface. The mycelia samples were stored at −70 °C until RNA extraction.

Total RNA was extracted from samples using the RNeasy kit (Qiagen, Valencia, CA, USA) in accordance with the manufacturer’s protocols. The purity of RNA was assessed by measuring the ratio of absorbances at 260 and 280 nm using a NanoDrop 1000 spectrophotometer (Thermo Fisher Scientific, Waltham, MA, USA), and the qualitative analysis of total RNA was performed by gel electrophoresis. Subsequently, cDNA synthesis was conducted using 2.0 μg total RNA, oligo (dT) 18 primer, random hexamers, and SuperScript III Reverse Transcriptase (Invitrogen, San Diego, CA, USA) following the instructions of the manufacturer.

Quantitative real-time PCR (qRT-PCR) was performed using a Real-Time PCR System (Bio-Rad) to analyze the transcription profiles of four genes involved in the DON biosynthesis cluster (tri4, tri5, tri6, tri10) and the housekeeping gene β-tubulin as an internal control. Primer sequences are listed in Appendix A. Each qRT-PCR reaction was performed in a 20 µL volume containing 1.5 µL cDNA template, 12.5 μL SYBR Green Super mix (Bio-Rad, Hercules, CA, USA), and 100 nM primers with nuclease-free water added to adjust the total volume. The thermal cycling conditions consisted of an initial denaturation at 95 °C for 5 min, followed by 40 cycles of 94 °C for 30 s, 57 °C for 30 s, and 72 °C for 15 s. The relative quantification of gene expression was established by using the 2^−ΔΔCt^ method. Statistical analysis was performed by using *t*-test.

### 2.6. Effect of Volatiles of Trichoderma spp. on F. graminearum Growth and DON/3-ADON/15-ADON Production

Split plates were used to assess the impact of volatiles emitted by *Trichoderma* on *F. graminearum* PG-Fg1 growth. PDA media were prepared and poured into 9 cm diameter split plates, with 10 mL dispensed on each side. After solidification of the media, a 6 mm mycelial plug from the edge of 7-day-old colony of PG-Fg1 was placed 1 cm from the edge of the split plate. A 5.5 cm diameter Petri dish inoculated with *Trichoderma* was allowed to grow for 3 days until it was fully covered, after which it was placed on the opposite side of the split plate. The colonies of PG-Fg1 were incubated at 28 °C in the dark for 5 days. Control dishes were prepared by inoculating PG-Fg1 on PDA. The experiment was carried out in triplicate.

The colony diameter of PG-Fg1 was measured with a ruler under a dissecting microscope every day. The growth inhibition of PG-Fg1 caused by *Trichoderma* volatiles was calculated with the formula%I_c_ = (C_C_ − C_T_)/C_C_ × 100(3)
where C_T_ was the average diameter of PG-Fg1 colonies confronted with *Trichoderma* volatiles, and C_C_ was the average diameter of PG-Fg1 colonies grown in control plates.

After 5 days of incubation of PG-Fg1, the experiment was made as described in Section 2.3 to analyze the production of DON, 3-ADON, or 15-ADON. The inhibition rate of DON, 3-ADON, and 15-ADON production was calculated with the formula %Ia.

### 2.7. The Tolerance of Trichoderma spp. to the Toxicity of F. graminearum Metabolites

The inhibitory effects of the culture supernatant of *F. graminearum* PG-Fg1 on *Trichoderma* spp. growth were studied in PDA. The PDA media were prepared and sterilized. After cooling but not solidifying, 15 mL of the medium was mixed with 5 mL culture supernatant of PG-Fg1 growing 5 days in PDB. The mixture was thoroughly mixed using a vortex mixer and quickly poured into 9 cm diameter Petri dishes. After the medium solidification, a 6 mm mycelial plug of *Trichoderma* was laid on the center of the Petri dish. The cultures were incubated at 28 °C in the dark for 7 days. Control dishes were prepared by inoculating *Trichoderma* on standard PDA without *F. graminearum* culture supernatant. The experiment was carried out in triplicate.

The colony diameter of *Trichoderma* was measured with a ruler under a dissecting microscope every day. The growth inhibition of *Trichoderma* (%I_d_) caused by *F. graminearum* PG-Fg1 culture supernatant was calculated with the following formula%I_d_ = (D_C_ − D_T_)/D_C_ × 100(4)
where D_T_ was the average diameter of *Trichoderma* colonies grown on media mixed with PG-Fg1 culture supernatant, and D_C_ was the average diameter of *Trichoderma* colonies grown in the control.

### 2.8. Potential of Trichoderma spp. to Produce DON/3-ADON/15-ADON

The ability of 28 strains to produce DON, 3-ADON, and 15-ADON was tested in PDB. For each *Trichoderma* strain, five 6 mm diameter mycelial plugs were added to 100 mL of PDB in a flask and incubated for 7 days with constant shaking (175 rpm) at 28 ± 1 °C. The culturing broth was collected, centrifuged, filtered, and used to analyze the concentration of DON, 3-ADON, and 15-ADON with a method of UPLC-MS/MS. The experiment was carried out in triplicate.

### 2.9. Determination of DON/3-ADON/15-ADON by UPLC-MS/MS

Mycotoxins were determined using a UPLC-MS/MS platform. Chromatographic separations were performed on a Waters Acquity-System (Milford, MA, USA). The column used for LC separations was a 100 mm × 2.1 mm i.d., 1.7-μm, Acquity UPLC BEH C18, equipped with an Acquity UPLC column in-line filter (0.2-μm). Sample and column temperatures were set at 15 °C and 35 °C, respectively. The mobile phase consisted of eluent A (deionized water containing 0.1% formic acid) and eluent B (methanol). A binary gradient with a flow rate of 0.3 mL/min was programmed.

MS/MS analyses were performed on a triple quadrupole mass spectrometer (QTRAP 5500 with an electrospray ionization (ESI) source, AB Sciex, Framingham, MA, United States) in a positive MRM mode. The mass parameters were as follows: 5.5 kV ion spray voltage, 20 psi curtain gas pressure, 60 psi pressure for the nebulizer (gas 1) and turbo (gas 2) gases, and 550 °C turbo heater temperature. For targeted quantitative analysis of mycotoxins, including ion confirmation using one quantifier and one qualifier transition, the transitions monitored (corresponding collision energy) were as follows: DON: 297.2 → 249.1 (16 V); 297 → 203.2 (21 V); 3-ADON: 339.0 → 231.0 (15 V); 339.0 → 213.0 (21 V); 15-ADON: 339.0 → 137.0 (15 V); 339.0 → 261.0 (18 V); the declustering potential voltage was set to 80 V. With this condition, the retention time of ADON, 3-ADON, and 15-ADON was about 3.1 min, 4.1, and 4.4 min, respectively. The method has a limit of quantification (LOQ) of 10.0 ng/mL based on a signal-to-noise ratio of 10:1.

### 2.10. Statistical Analysis

Statistical analyses were conducted using one-way analysis of variance (ANOVA) followed by Tukey–Kramer multiple comparison tests. All statistical analyses were conducted using GraphPad Instat 3.0 software (GraphPad Software, San Diego, CA, USA). The normality assumption for all continuous variables was formally evaluated using Shapiro–Wilk tests (α = 0.05). Where data distributions violated the normality assumption (e.g., [mention specific dataset/variable]), and transformations (e.g., [log/square root]) failed to achieve normality, non-parametric equivalents were employed. Figures were generated using OriginPro 9.0 software (OriginLab Corporation, Northampton, MA, USA).

## 3. Results

### 3.1. Interactions Between Trichoderma spp. and F. graminearum in Dual-Culture Assay

#### 3.1.1. *Trichoderma* spp. Were Antagonistic Against *F. graminearum* Growth in Dual-Culture

As shown in Appendix A, the daily radial growth of *Trichoderma* strains ranged from 7.7 to 19 mm/day, whereas *F. graminearum* PG-Fg1 exhibited an average daily radial growth rate of 6.5 mm/day, considerably slower than most *Trichoderma* strains. Rapidly growing *Trichoderma* strains effectively encircled and inhibited further expansion of the PG-Fg1 colony. In several confrontations, early contact between *Trichoderma* strains and PG-Fg1 led to the cessation of PG-Fg1 growth. After 21 days of co-culture (Figure 1), a more comprehensive merit of the antagonistic potential of *Trichoderma* strains against PG-Fg1 was observed. The 13/28 *Trichoderma* strains(GC-T2, GC-T4, GC-T24, GC-T27, GC-T76-1, GC-T40, GC-T40-1, GC-T41, GC-T75-1, GC-T82, GC-T83, GC-T85, GC-T88) arrested the growth of PG-Fg1 after contact and overgrew the PG-Fg1 colony (type 1). The *Trichoderma* strains GC-T20, GC-T26, and GC-T18-1 aggressively overgrew the PG-Fg1 colony, but PG-Fg1still grew (type 1/2). The remaining *Trichoderma* strains exhibited less aggressive interactions, characterized by mutual inhibition zones ≤ 2 mm wide. We found that three *Trichoderma* strains, including *T. asperellum* GC-T2, GC-T4, and *T. harzianum* GC-T40, were able to aggressively overgrow and sporulate on the colony of PG-Fg1, indicating these three *Trichoderma* strains are more effective in inhibiting the mycelia spread of *Fusarium*. Conversely, the pathogen strain PG-Fg1 exhibited limited ability to overgrow *Trichoderma* for further expansion.

#### 3.1.2. *Trichoderma* spp. Inhibit DON/3-ADON/15-ADON Production by *F. graminearum* in Dual-Culture

The strain *F. graminearum* PG-Fg1 produced DON, 3-ADON, and 15-ADON at concentrations of 2.15 ± 0.60, 11.33 ± 4.03, and 4.03 ± 1.48 μg/g, respectively, after 7 days of incubation on PDA. When PG-Fg1 was co-cultured in dual-culture with *Trichoderma*, the amounts of DON, 3-ADON, and 15-ADON produced by *F. graminearum* significantly reduced due to the strong antagonistic effect of *Trichoderma* (Figure 2). Data showed that the 21/28 *Trichoderma* strains were effective in reducing the production of DON, 3-ADON, and 15-ADON. With regard to DON, the antagonists exhibited stronger inhibitory capacity: the inhibition rate of DON production was significantly higher than 3-ADON and 15-ADON production when directly confronting *Trichoderma* strains GC-T26, GC-T40, GC-T43-1, and GC-T83. Interestingly, strains GC-T75-1 and GC-T45-1 increased the production of DON, 3-ADON, and 15-ADON, though statistically non-significantly in the case of GC-T45-1. Therefore, it is essential to monitor DON and its acetylated derivatives when assessing the anti-toxigenic activity of antagonistic *Trichoderma* strains in dual-culture.

### 3.2. Influence of Trichoderma Metabolites on F. graminearum

#### 3.2.1. *Trichoderma* Non-Volatile Metabolites Inhibit *F. graminearum* Growth

The inhibitory effects of non-volatile metabolites produced by the 28 *Trichoderma* strains were investigated on *F. graminearum* PG-Fg1 growth on PDA. Figure 3A shows the PG-Fg1 colonies growing in the presence of non-volatiles of some *Trichoderma* strains. The 23/28 *Trichoderma* strains significantly inhibited *F. graminearum* growth, with inhibition percentages ranging from 7 ± 1% to 72 ± 6% (Figure 3B). *T. gamsii* GC-T26 was the most effective strain, resulting in 72 ± 6% growth inhibition. The more effective strains were *T. atroviride* GC-T78, achieving over 60% inhibition on PDA.

#### 3.2.2. *Trichoderma* Non-Volatile Metabolites Inhibit DON/3-ADON/15-ADON Production

The inhibitory effects of *Trichoderma* non-volatiles on DON, 3-ADON, and 15-ADON production are shown in Figure 4A. Except for GC-T8 and GC-T88, all other *Trichoderma* strains significantly reduced the production of DON, 3-ADON, and 15-ADON. The reduction rates ranged from 4 to 98%, 9 to 98%, and 13 to 98% for DON, 3-ADON, and 15-ADON, respectively. Notably, 10 of the 28 strains were more effective in the reducing of DON, 3-ADON, and 15-ADON production by PG-Fg1, with average reduction ratios over 80%. Specifically, *T. gamsii* GC-T26, the most potent PG-Fg1 growth inhibitor (Figure 3), achieved reductions of 98%, 98%, and 97% in DON, 3-ADON, and 15-ADON production, respectively.

Interestingly, non-volatiles of strains GC-T8 and GC-T88 increased the production of DON, 3-ADON, and 15-ADON, despite significantly inhibiting PG-Fg1 growth. In order to investigate the molecular mechanism of *Trichoderma* GC-T88 increasing the mycotoxin production, the expression levels of DON biosynthesis genes (*tri4*, *tri5*, *tri6*, *tri10*) were studied in PG-Fg1 grown on PDA premixed with GC-T88 culture supernatant. As shown in Figure 4B, in the presence of GC-T88 metabolites, the expression levels of tri4 and tri5 in PG-Fg1 were higher than those in controls. Conversely, the two genes tri6 and tri10 remained similar in expression level to those in the control. Therefore, non-volatiles of *Trichoderma* GC-T88 up-regulated the expression of DON biosynthesis genes of tri4 and tri5.

#### 3.2.3. *Trichoderma* Volatile Metabolites Inhibit *F. graminearum* Growth

Split plates were used to investigate the influence of volatiles emitted by *Trichoderma* spp. on *F. graminearum* PG-Fg1 (Figure 5A). Figure 5B shows the inhibitory effects of *Trichoderma* volatile metabolites on *F. graminearum* PG-Fg1 growth. Volatiles from six *Trichoderma* strains significantly inhibited PG-Fg1 growth, with the inhibition ratios ranging from 16.7% to 31.8%. *T. viride* GC-T45-1 was the most inhibitory strain to PG-Fg1 growth, and almost similar efficiency was observed for *T. gamsii* GC-T26. Other *Trichoderma* strains also showed inhibitory effects on PG-Fg1 growth, but the inhibitory effect was less potent.

#### 3.2.4. *Trichoderma* Volatile Metabolites Inhibit DON/3-ADON/15-ADON Production by *F. graminearum*, While Volatiles of GC-T45-1 Act Oppositely

The inhibitory effects of *Trichoderma* volatiles on DON, 3-ADON, and 15-ADON production are shown in Figure 6A. Most *Trichoderma* strains (21 out of 28) significantly inhibited DON production. Interestingly, the volatiles of GC-T45-1 increased the production of all three mycotoxins (DON, 3-ADON, and 15-ADON). Indeed, GC-T45-1 volatiles did significantly inhibit the growth of PG-Fg1 (Figure 5B).

In order to investigate the molecular mechanism, we analyzed expression levels of DON biosynthesis genes (*tri4*, *tri5*, *tri6*, *tri10*) in PG-Fg1 exposed to GC-T45-1 volatiles. As shown in Figure 6B, *tri4* expression in PG-Fg1 was significantly higher when exposed to GC-T45-1 volatiles compared to controls. Conversely, no statistically significant differences were observed in the expression levels of the *tri5*, *tri6*, and *tri10* genes between the treatment and controls.

### 3.3. Interactions Between DON/3-ADON/15-ADON and Trichoderma

#### 3.3.1. *Trichoderma* Tolerance to the Toxicity of *F. graminearum* PG-Fg1 Metabolites

The strain *F. graminearum* PG-Fg1 produced DON, 3-ADON, and 15-ADON at concentrations of 132 ± 12, 209 ± 9, and 164 ± 11 ng/L, respectively, after 7 days of incubation in PDB. However, the culture supernatant of PG-Fg1 did not significantly affect the growth of most *Trichoderma* strains (Figure 7), suggesting that these strains exhibit tolerance to the toxicity of DON, 3-ADON, and 15-ADON. Surprisingly, our experimental results showed that the culture supernatant of PG-Fg1 promoted the growth of certain *Trichoderma* strains. In particular, it significantly promoted the growth of *Trichoderma* strains GC-T40, GC-T43-1, and GC-T45-1, indicating that *Trichoderma* can use *F. graminearum* metabolites as nutrients for its own growth. Further analysis of stimulating component(s) may be promising.

#### 3.3.2. *Trichoderma* spp. Have No Ability to Produce DON/3-ADON/15-ADON

All 28 *Trichoderma* strains were cultured in PDB for 7 days. Then, DON, 3-ADON, and 15-ADON in the culture supernatants were analyzed. The results showed that none of the strains could produce DON, 3-ADON, or 15-ADON.

## 4. Discussion

*Fusarium* species, a prominent genus of plant pathogenic fungi, target various plant parts such as grains, roots, seedlings, stems, and heads. These fungi are major producers of mycotoxins including fumonisins, trichothecenes, and zearalenone. Mycotoxin contamination of agricultural products poses a significant challenge to global food safety and trade. Effective mycotoxin management strategies include pre-harvest control of *Fusarium* infections and post-harvest mycotoxin removal. *Trichoderma* strains demonstrate superior abilities to prevent pathogen infections and reduce pre-harvest fungal disease incidence through competition, mycoparasitism, antagonism, and antibiosis [28].

However, besides the relationship between *Trichoderma* and the pathogenic fungus *F. graminearum*, our present study attempted to address the interactions between *Trichoderma* and trichothecenes biosynthesis. We found that the antagonistic efficacy of *Trichoderma* depends on both *Fusarium-Trichoderma* strain combinations and their effects on trichothecene biosynthesis. The dual-culture confrontation test, widely used to study the interaction between *Trichoderma* and various pathogens, has also been utilized for picking strains that exhibit antagonistic properties against these pathogens [32,43,44,45]. These biological control agents are being intensively studied as potential replacements for chemical fungicides [46,47]. In our confrontation tests, the daily growth and aggressive interactions between *Trichoderma* strains and *F. graminearum* PG-Fg1 were revealed. However, it became evident that the interaction patterns observed in dual-culture assays were insufficient to fully assess the antagonistic efficacy of *Trichoderma* strains. For instance, *Trichoderma* strains GC-T75-1 and GC-T45-1 exhibited behaviors such as surrounding and overgrowing the *F. graminearum* colony, yet significantly increased the production of DON, 3-ADON, and 15-ADON. Therefore, it is imperative to evaluate mycotoxin production to accurately determine the potential antagonistic capabilities of *Trichoderma* strains, because some strains such as GC-T75-1 and GC-T45-1 promoted *F. graminearum* production of DON, 3-ADON, and 15-ADON when directly confronting. Based on our experimental results, we observed the following: increasing DON levels were observed in dual-culture assays withGC-T45-1andGC-T75-1; non-volatile compounds produced byGC-T8andGC-T88increased DON levels; volatile compounds produced byGC-T45-1increased DON levels. First, the DON-enhancing effect ofGC-T45-1in dual-culture aligns with its volatile compounds’ capacity to increase DON, indicating that its volatiles drive the elevated toxin levels in co-culture. However, GC-T75-1 exhibits complex behavior: while it increased DON in dual-culture, its non-volatile compounds significantly reduced DON levels, and its volatiles showed no significant effect. We therefore postulate that the DON increase byGC-T75-1in dual-culture is also attributable to its volatile compounds. However, subtle changes in fungal interactions and growth conditions (e.g., pH, temperature, nutrients, duration) during dual-culture could alter volatile compound production/activity. Additionally, physical hyphal contact betweenGC-T75-1and *Fusarium graminearum* PG-Fg1 in dual-culture may trigger stress responses that enhance DON production. While non-volatile compounds fromGC-T8andGC-T88increased DON in isolation, no such increase occurred in dual-culture. We hypothesize this is because DON measurements were taken after only 7 days of co-culture—when hyphal contact was minimal or absent—preventing sufficient exposure to the non-volatile compounds for them to exert their effects.

Usually, the capability of *Trichoderma* against pathogens is determined by measuring the percentage inhibition of pathogen growth. For example, *T. asperellum* ZJSX5003 has been demonstrated producing peptaibols and polyketides against *F. graminearum* growth [44]. This strain reduced the incidence of corn stalk rot by 71%, indicating that the *T. asperellum* strain ZJSX5003 is a potential source for the development of a biocontrol agent against corn stalk rot. Our study employed bioassays to evaluate *Trichoderma*’s inhibitory effects on both *F. graminearum* growth and mycotoxin production. In total, 2 of 28 *Trichoderma* strains (GC-T8 and GC-T88) produced non-volatiles, significantly increasing the production of DON, 3-ADON, and 15-ADON by *F. graminearum* PG-Fg1, despite significantly reducing its growth. Similarly, 1 of the 28 *Trichoderma* strains tested, namely GC-T45-1, could produce volatiles which significantly increased the production of DON, 3-ADON, and 15-ADON by PG-Fg1. In our previous study [48], we observed a similar phenomenon that the *Trichoderma* strains ITEM 4484 and T8 increased the production of AfB1 by *A. flavus* Af-9, despite inhibiting its growth. Stracquadanio et al. also reported a similar case that 0.78 mg/mL *T. asperellum* IMI393899 extract increased ochratoxin A (OTA) production by *P. verrucosum* [49]. We explored the possible underlying mechanism in our trials. We found an increase in the expression of *tri4* and *tri5* of the trichothecene gene cluster in the presence of GC-T88 non-volatiles, as well as an upregulation of *tri4* in the presence of GC-T45-1 volatiles. Although further analyses are necessary to examine whether the expression of other genes of the biosynthesis cluster may be affected, it can be suggested that some metabolites produced by *Trichoderma* could activate a signaling pathway of *Fusarium*, leading to the upregulation of mycotoxin biosynthesis genes. Similarly to certain fungicidal substances, these metabolites can stimulate the production of mycotoxins because of induced stress. For instance, the application of fungicide azoxystrobin in wheat contaminated with *Fusarium* spp. increased the production of DON [50]. Therefore, it was suggested that *Fusarium* spp. can upregulate DON biosynthesis genes to produce more DON as weapons to resist *Trichoderma*, probably due to the stress induced by the *Trichoderma* metabolites. Therefore, this study underscores the potential hidden hazardous effects of *Trichoderma* as biocontrol agents that can stimulate toxigenic fungi (e.g., *Fusarium*) to produce more mycotoxins, which will contribute to the proper selection of potential biocontrol agents.

In our study, in addition to underscoring the potential hazardous effects of *Trichoderma* as antimicrobial agents, we also evaluated the antagonistic abilities of *Trichoderma* against *F. graminearum*. *Trichoderma* grew faster, rapidly spread, and surrounded the colony of *F. graminearum* to stop its further enlargement; both non-volatiles and volatiles of most strains could significantly inhibit *F. graminearum* growth and DON production. Moreover, we observed that *F. graminearum* PG-Fg1 showed no inhibitory effects on *Trichoderma* growth, but significantly promoted the growth of 3 out of the 28 *Trichoderma* strains, suggesting *Trichoderma* possesses resistance to mycotoxin toxicity and can utilize pathogen initial metabolites as nutrients to stimulate its own growth. These results highlight the potential of *Trichoderma* to control toxigenic fungi and mycotoxin contamination. Previous reports also demonstrated the potential of *Trichoderma* as biocontrol agents. For instance, the antagonistic *Trichoderma* strains, once spread on the soil surface, gradually displace the toxigenic fungi by mechanisms of competition and/or mycoparasitism [30,31,44]. Enzymes [51], peptaibols [52], polyketides [53], sesquiterpenes [54], antimicrobial peptides [55], and other metabolites [56] purified from fungi *Trichoderma* have been shown to exhibit strong inhibitory effects on the growth and mycotoxin biosynthesis of pathogenic fungi, including *Penicillium*, *Fusarium*, and *Colletotrichum*. Regarding the interaction between *Trichoderma* and mycotoxins, several *Trichoderma* species have been demonstrated to have the ability to degrade aflatoxins [41], zearalenone [57], ochratoxin A [58,59], and trichothecenes [60,61]. Therefore, our experimental results align with the general conclusions that *Trichoderma* could antagonistically compete for nutrients and vivosphere, and parasitize the challenged pathogens by mechanisms of competition and mycoparasitism, and release non-volatiles and volatiles as weapons to biologically control toxigenic pathogens and mycotoxin biosynthesis in indirect fungal interaction.

## 5. Conclusions

In conclusion, the findings presented herein have shown the multimodal interactions of *Trichoderma* with toxigenic *F. graminearum* and its trichothecenes DON/3-ADON/15-ADON. Despite the potential of *Trichoderma* strains to be microbial agents in agriculture, their hazardous effects are poorly understood compared to their biocontrol abilities. The in vitro selection of effective strains for further assessment should be conducted systematically, considering the stimulatory effect on mycotoxins production. Metabolites produced by some *Trichoderma* strains could trigger increased mycotoxin production, indirectly acting as accomplices to pathogenicity and inducing stress. Therefore, strain characterization and selection must be undertaken with caution.

Nonetheless, most *Trichoderma* strains, as antagonists, can compete for nutrients and virosphere, parasitize target pathogens through competition and mycoparasitism, and release non-volatiles and volatiles as weapons to biocontrol toxigenic pathogens and mycotoxin biosynthesis. Therefore, *Trichoderma* spp. remain promising candidates and viable alternatives to existing biocontrol agents, with effective biocontrol strategies potentially leveraging multiple strains with diverse mechanisms of action. However, a thorough understanding of the interactions between toxigenic pathogens, mycotoxins, and *Trichoderma* is essential to avoid a more serious mycotoxin contamination resulting from the indiscriminate use of microbial agents.

## Figures and Tables

**Figure 1 jof-11-00521-f001:**
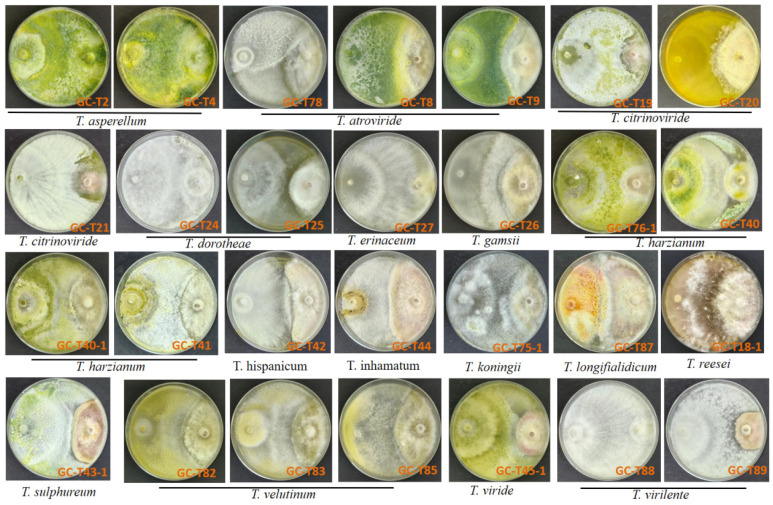
Dual-culture test of *Trichoderma* spp. (on the left-hand side of the Petri dishes) and *F. graminearum* PG-Fg1 (on the right-hand side) colonies on PDA after 21-day growth at 25 °C.

**Figure 2 jof-11-00521-f002:**
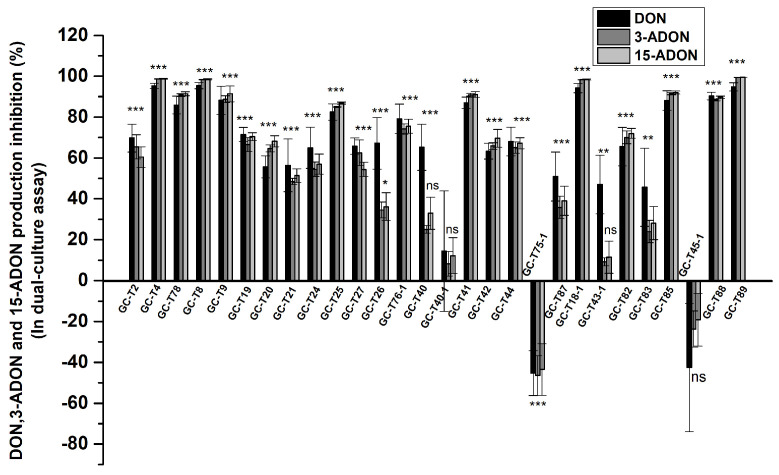
Inhibition percentage of DON, 3-ADON, and 15-ADON production by *F. graminearum* PG-Fg1 grown for 7 days on PDA when confronting *Trichoderma* spp. in dual-culture, as compared to control. Values are the means ± SD (n = 3). Asterisks indicate statistically significant differences with control group at *p* > 0.05 (ns), *p* < 0.05 (*), *p* < 0.01 (**), or *p* < 0.001 (***) calculated by one-way ANOVA.

**Figure 3 jof-11-00521-f003:**
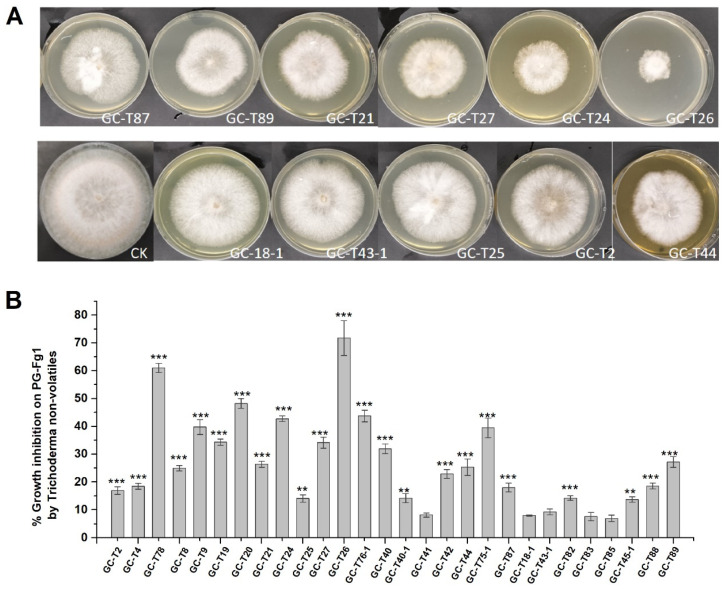
(**A**) *F. graminearum* PG-Fg1 colonies growing in the presence of *Trichoderma* non-volatiles. (**B**) Inhibitory effect of *Trichoderma* spp. non-volatiles on the growth of PG-Fg1. Values are the means ± SD (n = 3) of the percent reduction in colony diameter as compared to control. Asterisks indicate statistically significant differences with control group at *p* < 0.05, *p* < 0.01 (**) or *p* < 0.001 (***) calculated by one-way ANOVA.

**Figure 4 jof-11-00521-f004:**
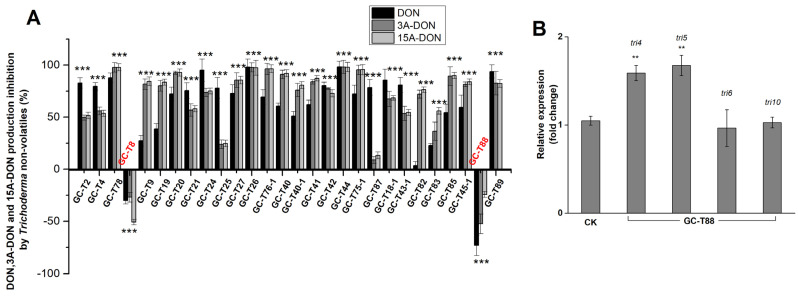
(**A**) Percent reduction in DON, 3-ADON, and 15-ADON production of *F. graminearum* PG-Fg1 grown for 7 days on PDA containing *Trichoderma* non-volatiles, as compared to control. Values are the means ± SD (n = 3). Asterisks indicate statistically significant differences with control group at *p* < 0.05 or *p* < 0.001 (***) calculated by one-way ANOVA. (**B**) Expression analyses of DON biosynthesis genes in PG-Fg1 after 5 days of growth on medium without (control, ck) and with GC-T88 non-volatiles. The *β-tubulin* gene was used as reference gene. Value of gene expression with statistically significant differences to control (*p* ≤ 0.01) is indicated by asterisks (**).

**Figure 5 jof-11-00521-f005:**
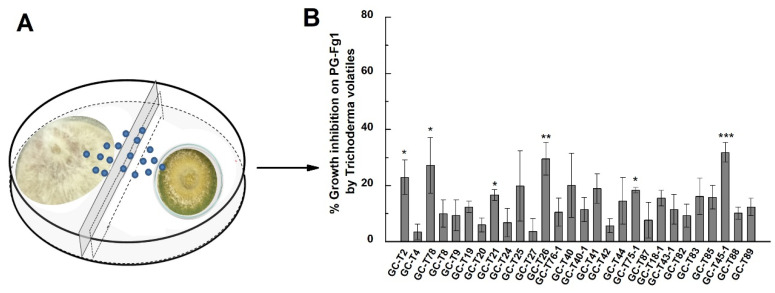
(**A**) The system of split plates employed to investigate the interaction of volatiles emitted by *Trichoderma* spp. (on the right-hand side) with *F. graminearum* PG-Fg1 (on the left-hand side). (**B**) The inhibition on *F. graminearum* PG-Fg1 growth by volatiles of *Trichoderma* strains. Values are the means ± SD (n = 3). Asterisks indicate statistically significant differences with control group at *p* < 0.05 (*), *p* < 0.01 (**), or *p* < 0.001 (***) calculated by one-way ANOVA.

**Figure 6 jof-11-00521-f006:**
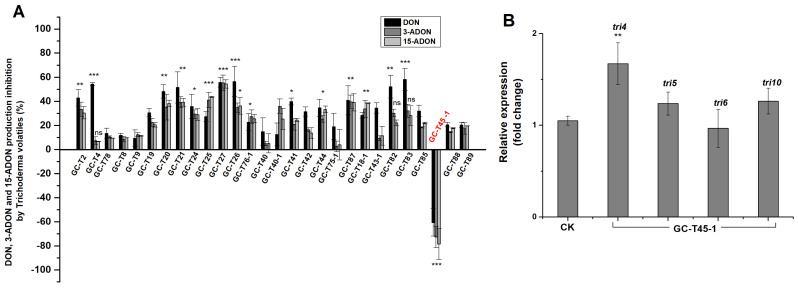
(**A**) Percent inhibition of DON, 3-ADON, and 15-ADON production by *F. graminearum* PG-Fg1 in the presence of *Trichoderma* volatiles, as compared to control. Values are the means ± SD (n = 3). Asterisks indicate statistically significant differences with control group at *p* > 0.05 (ns), *p* < 0.05 (*), *p* < 0.01 (**), or *p* < 0.001 (***) calculated by one-way ANOVA. (**B**) Expression analyses of DON biosynthesis genes in PG-Fg1 after 5 days of growth on medium without (control) and with GC-T45 non-volatile metabolites. Value of gene expression with statistically significant differences to control (*p* ≤ 0.01) is indicated by asterisks (**).

**Figure 7 jof-11-00521-f007:**
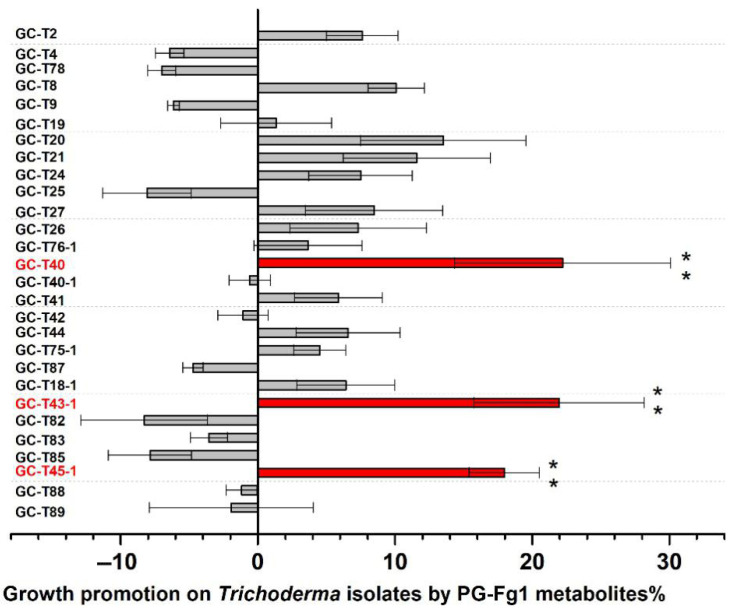
The promotion of *Trichoderma* growth by metabolites of *F. graminearum* PG-Fg1. Values are the means ± SD (n = 3). Asterisks indicate statistically significant differences with control group at *p* < 0.01 (**) calculated by one-way ANOVA.

## Data Availability

The original contributions presented in this study are included in the article/Appendix A. Further inquiries can be directed to the corresponding authors.

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
