# Peer review of "Interaction of Trichoderma Species with Fusarium graminearum Growth and Its Trichothecene Biosynthesis as Further Contribution in Selection of Potential Biocontrol Agents"

_jof, 2025, doi:10.3390/jof11070521_

Round 1

Reviewer 1 Report

Dear authors,

The subject of this manuscript is highly relevant, particularly given the increasing interest in the dual role of Trichoderma spp. in plant protection and its potential unintended effects on food safety. The authors address an important and relatively underexplored topic: the influence of both volatile and non-volatile metabolites of Trichoderma on mycotoxin production of Fusarium graminearum. Their key finding—that certain Trichoderma isolates may enhance mycotoxin production despite exhibiting antifungal activity—is particularly noteworthy and holds significant implications for biocontrol applications and food safety risk assessments.

The major methodological concern refers to the molecular identification of Trichoderma isolates. The use of the ITS region alone is insufficient for accurate species-level identification within this genus. It is strongly recommended that the authors include additional phylogenetic markers, such as EF1-α or RPB2, to enhance the taxonomic resolution and reliability of the identification. Moreover, the inclusion of accession numbers for the sequences submitted to public databases (e.g., NCBI GenBank) would strengthen the transparency and reproducibility of the study.

Regarding the interpretation of the results, there remains some ambiguity that should be addressed in the Discussion section. Specifically, it is unclear why isolates GC-T45-1 and GC-T75-1 increased mycotoxin production in the dual culture (DC) assay (3.1.2) but not in the assay involving non-volatile metabolites (3.2.2). For isolate GC-T45-1, the observed increase in mycotoxin production in the volatile compound assay (3.2.4), combined with its inhibitory effect on fungal growth (3.2.3), suggests that its mode of action is primarily through volatile metabolites. However, the behavior of isolate GC-T75-1 is more perplexing: despite good inhibition of F. graminearum in the volatile assay (3.2.3), no increase in mycotoxin production was observed (3.2.4). Similarly, isolates GC-T8 and GC-T88 significantly increased toxin levels in the non-volatile assay (3.2.2), yet this effect was not reflected in the DC test. Given that the activity of these isolates appears to be mediated by non-volatile metabolites, one would expect at least some corresponding effect in the DC assay, even at lower concentrations.

You should consider providing a more detailed discussion or hypothesis to account for these inconsistencies, possibly related to metabolite stability  or interactions in the dual culture environment. Such elaboration would enhance the scientific robustness and interpretative clarity of the manuscript.

Minor technical suggestions have been provided in the attached PDF file for the authors’ consideration.

Reviewer 2 Report

This manuscript presents a relevant and well-executed study on the interactions between Trichoderma spp. and Fusarium graminearum, with a focus on growth inhibition and mycotoxin suppression. The experimental design is robust and includes diverse approaches, such as dual-culture assays, metabolite analysis, and gene expression profiling. The findings are novel and provide valuable insights into both the benefits and potential risks of Trichoderma use as biocontrol agents. With minor revisions, the manuscript is suitable for publication in Journal of Fungi.

Title

  • The title should be more concise and accurately reflect the main focus of the manuscript.

Abstract

  • The abstract is overly long and lacks a clear structure. It should be revised to briefly and objectively present the study's objectives, materials and methods, key results, and main conclusions.

Introduction

  • Line 4: Is this citation format consistent with the journal's style? All references in the format “(Author, year)” should be replaced with numbered citations in square brackets, following the order of appearance in the text.
  • Line 41: Scientific names must be italicized.
  • Line 49: Italicize the scientific name.
  • Line 81: Citation should be adjusted to match the journal's style.

Materials and Methods

  • Lines 122–125: Please specify the DNA extraction protocol or kit used. What sequencing platform was employed? Confirm whether identification was performed only at the genus level, as ITS alone does not reliably resolve species-level identification. Are GenBank accession numbers available?
  • Lines 127–128: Was this the only method used to preserve fungal isolates? Clarify if any other preservation methods were applied.
  • Lines 150–151: This procedure requires more detail. Please cite references that support or describe the method used.
  • Lines 168–169: Was a concentration of 25% v/v used? Was the method based on Dennis and Webster? Please clarify and cite accordingly.
  • Line 208: The referenced table is not accessible via the provided link or on the Journal of Fungi
  • Line 228: A supplementary file is mentioned, but it is not available for review. Ensure all supplementary materials are properly uploaded.
  • Lines 277–282: Were assumptions of normality assessed for statistical analyses? If data were not normally distributed, were transformations applied? These details should be explicitly stated.

References

  • References must be formatted according to MDPI's guidelines (numerical style, with full citation details and DOI where available).
  • Scientific names should be italicized in this section as well.

Round 2

Reviewer 1 Report

Dear Authors,

The revised version of the manuscript represents a substantial improvement. While the identification of Trichoderma species based primarily on morphological characteristics and ITS sequencing may not represent the most robust or current methodology, the topic remains of significant interest, and the presented results are compelling. Considering that study does not deeply explore intra- or interspecific diversity within Trichoderma, I consider the identification of Trichoderma species acceptable in its current form.

I have two minor suggestions for further improvement:

  1. The added section (currently located on lines 475–492) would be more logically positioned after the section ending on line 443 to enhance the coherence of the discussion.
  2. I recommend revising the title of the supplementary files to:
    "Multimodal interactions of toxigenic Fusarium graminearum and deoxynivalenol with Trichoderma indicate potential effects of Trichoderma as a microbial agent."

Author Response

NOTE: The words in blue are comments or questions from reviewers and editors, and the words in black are our responses.

Dear reviewer,

We would like to take this opportunity to express our sincere gratitude for your time and efforts inreviewing our article. We are truly honored to have the benefit of your expertise and insights. We read the manuscript and all comments very carefully. We have tried to revise our manuscript according to all suggestions. The changes are highlighted with red text. Point-by-point responses to reviewers are listed on the review frame.

Thank you very much again for your help.

Responds to the reviewer’s comments:

  1. The added section (currently located on lines 475–492) would be more logically positioned after the section ending on line 443 to enhance the coherence of the discussion.

Response:

Thank you for your professional suggestions. We fully agree with your suggestion and have implemented the recommended change as highlighted in red in the revised manuscript.

  1. I recommend revising the title of the supplementary files to:

"Multimodal interactions of toxigenic Fusarium graminearum and deoxynivalenol with Trichoderma indicate potential effects of Trichoderma as a microbial agent."
Response:

We are grateful for your expert and valuable comments. We fully accept your suggestion and have accordingly revised the titles of the supplementary files in our updated submission.